# Recovery of Diamond and Cobalt Powders from Polycrystalline Drawing Die Blanks via Ultrasound Assisted Leaching Process—Part 2: Kinetics and Mechanisms

**Ferdinand Kießling [1], Srecko Stopic [2,\*], Sebahattin Gürmen [3] and Bernd Friedrich [2]**

[1] Redies Deutschland GmbH & Co.KG, Metzgerstr. 1, 52070 Aachen, Germany; fer87den@gmail.com
[2] IME Process Metallurgy and Metal Recycling, RWTH Aachen University, Intzestrasse 3, 52056 Aachen, Germany; bfriedrich@ime-aachen.de
[3] Metallurgical and Materials Engineering Department, Istanbul Technical University, Ayazaga Campus, Istanbul 34469, Turkey; gurmen@itu.edu.tr
\* Correspondence: sstopic@metallurgie.rwth-aachen.de; Tel.: +49-241-809-5860

**Abstract:** The leaching of industrial polycrystalline diamond (PCD) blanks in aqua regia at atmospheric pressure between 60 °C and 80 °C was performed using an ultrasound to improve the rate of cobalt removal in order to be able to reuse very expensive polycrystalline diamond. Because cobalt (20 wt.%) is used as a solvent catalyst in the production of PCD, its recovery is very important. The cleaned PCD are returned to the production process. Kinetic models were used in the study of cobalt dissolution from polycrystalline diamond blanks by measuring the declining ferromagnetic properties over time. For a better understanding of this leaching process, thermochemical aspects are included in this work. The lowest free Gibbs energy value was obtained with a low solid/liquid ratio and the full use of an ultrasound. A transition from a reaction-controlled to a diffusion-controlled shrinking core model was observed for PCD with a thickness greater than 2.8–3.4 mm. Intermittent ultrasound doubles the reaction rate constant, and the full use of ultrasound provides a 1.5-fold further increase. The obtained maximum activation energy between 60 °C and 80 °C is 20 kJ/mol, for a leaching of diamond blank with grain size of 5 μm.

**Keywords:** cobalt; aqua regia; polycrystalline diamond blanks; kinetics; thermochemistry

---

## Part 1: Experimental Design and Efficiencies

This study attempts to achieve optimal recovery of diamond and cobalt from polycrystalline diamond (PCD) blanks. In nine experimental runs of 5 days' duration, cobalt-containing PCD was leached in aqua regia at atmospheric pressure between 60 °C and 80 °C. Using two reactors in parallel, the temperature, ultrasound irradiation time, solid-to-liquid ratio, and PCD size were varied to find out which parameters are beneficial and could possibly accelerate the process. PCD weights and cobalt content in solution were monitored as well. It was found that aqua regia accumulated more dissolved cobalt at 60 °C than at 80 °C, probably due to volatile reagents being less available over time. With added ultrasound and at low S/L ratios, i.e., close to 15 g/L, the leaching time for D14 to reach a 90% leach mark was reduced to three days, a significant shortening. PCD type D18 with a thickness of 3.5 mm were not leached to completion after five days. Leaching temperature had more impact on the results than ultrasound. These findings were reinforced by the mass balance in which a small discrepancy was found. The PCD lost a fraction of weight that could not be explained by the weight of dissolved cobalt. From EDS (Energy Dispersive Spectroscopy) data and the nature of PCD,

this fraction probably consisted of, oxygen from oxides in the PCD, iron or single diamond grains that were broken off by the impact of the ultrasound.

## 1. Introduction

The term polycrystalline diamond (PCD) describes a variety of amorphous compounds mostly or wholly consisting of microscopically small diamond grains. A single crystal of natural diamond is anisotropic in terms of its mechanical and thermodynamic properties, including tensile strength and thermal conductivity, for instance. Most PCD will have a random arrangement of individual grains, resulting in a quasi-isotropic compound. However, there are forms of PCD that are made in a different way and that have different properties. Binderless PCD (Sumidia), CVD crystals [1], and monocrystalline dies in general will not be discussed herein. The conditions needed for diamond powder to form a framework are extreme. Only in the region of 50 kbar and at a temperature of 2000 °C will the desired reaction happen on reasonable time scales [2–4]. Cobalt is used as a solvent catalyst in the production of PCD; without it, the reaction would require even more pressure and a higher temperature.

The leaching solution in this case has the colloquial name "aqua regia" because it was found to dissolve noble metals such as gold or platinum; early records of its use date back centuries [5]. Aqua regia ensures an oxidation environment. More specifically, the aqua regia was mixed from 3 parts Merck KGaA fuming hydrochloric acid 37%, Emsure ACS/ISO quality and one part PanReac ApplicChem nitric acid 65% ISO analysis quality.

The solution is a mixture of hydrochloric acid, HCl, and nitric acid, $HNO_3$. Both are strong acids, and at a ratio of 3:1, reactions (1) to (3) occur [6].

$$3HCl_{(aq)} + HNO_{3(aq)} \leftrightarrow NOCl_{(aq)} + Cl_{2(g)} + 2H_2O_{(l)} \tag{1}$$

$$NOCl_{(aq)} + H_2O_{(l)} \leftrightarrow HNO_{2(aq)} + HCl_{(aq)} \tag{2}$$

$$2HNO_{2(aq)} \leftrightarrow NO_{(aq)} + NO_{2(aq)} + H_2O_{(l)} \tag{3}$$

The formation and transport of molecular chlorine gas and NOCl has been found to occur within minutes to hours [6]. Baghalha et al. [7] concluded that the 3:1 mixing ratio maximizes the production of chlorine per unit mass of reactants, and is to be favored when chlorine is the desired oxidizing agent. However, if the desired reaction requires only low pH or different oxidizing agents, this ratio or aqua regia itself may not be suitable. This study intends to extract cobalt from PCD as a chloride, $CoCl_{2(aq)}$, and therefore, uses aqua regia, or NOCl to be precise. The desired reaction in this case is the oxidation and dissolution of cobalt into aqua regia, which is achieved in the following redox-reactions:

$$Co_{metal} + 2NO^+_{aq} + 2Cl^-_{aq} \leftrightarrow Co^{2+}_{aq} + 2Cl^-_{aq} + 2NO_g \tag{4}$$

$$CoO_s + 2NO^+_{aq} + 2Cl^-_{aq} \leftrightarrow Co^{2+}_{aq} + 2Cl^-_{aq} + NO_g + NO_{2g} \tag{5}$$

The Pourbaix diagrams by Huang et al. [8] show that the equilibrium for this reaction should be on the right side of the balance, since the divalent cobalt cation is not only stable at pH << 1, but also at pH > 1. In many cases, the most cost- and energy-efficient way to extract metal from gangue or scraps is to oxidize and dissolve it in a leaching solution. There are many examples; the well-established Caron process is one of them [9]. It is used to treat lateritic nickel ores by reduction roasting and subsequent leaching for the purpose of obtaining a nickel-bearing solution while separating nickel from iron [10,11].

The polycrystalline diamond in aqua regia can be seen as a solid compound particle where its metallic components react with the solution. The reaction front moves inwards and leaves a layer of inert diamond grains behind. That is why the model of the shrinking unreacted core (SCM) is applied [12,13]. According to the model, the leaching rate may depend on the reaction or diffusion of

educts and products from the reaction site; in reality, it is often a mixture of both effects. If a linear relationship is found in the plots of Equations (6) and (7), over time, the model is confirmed and the apparent rate constant can be extracted from the slope. Equation (6) is the Ginstling-Brounshtein (D4) model. The D4 model is another type of diffusion three-dimensional model, in contrast to widely used Jander model, as reported by Khawan and Flanagan [14]. If a solid particle has a spherical or cubical shape, a contracting sphere/cube model can be applied, as shown in Equation (7).

$$1 - \frac{2}{3} X - (1 - X)^{\frac{2}{3}} = k_D \times t \tag{6}$$

$$1 - (1 - X)^{\frac{1}{3}} = k_R \times t \tag{7}$$

X is a dimensionless variable that represents the relative change in the amount of substance or concentration, which is why in leaching processes, the yield is taken for X, for example. In this study, k will be indexed with "D" or "R", depending on whether the diffusion- or chemical reaction-controlled model was applied.

From this apparent rate constant, the activation energy for the reaction can be obtained by plotting the natural logarithm of k over the reciprocal temperature in an Arrhenius plot, or by using the following equation [15]:

$$E_A = \partial \ln(k) / \frac{\partial}{T} \tag{8}$$

$$\Delta G^0 = -RT \ln(k) \tag{9}$$

The slope of the Arrhenius plot delivers the activation energy. Furthermore, the k-rate coefficient can be used to calculate the Gibbs energy accompanying this reaction using Equation (9)

The main aim of this work was to study the kinetics of cobalt removal from polycrystalline diamond blanks using an ultrasound-assisted leaching process; no reports of such a process exist in the literature. Two mathematical models will be tested in order to determine the activation energy and rate coefficient. An additional thermochemical analysis was included to provide a better explanation of the behavior of cobalt in a water solution at different pH-Eh values using an Eh-pH diagram.

## 2. Thermochemistry of Cobalt Leaching

A thermochemical analysis of Eh-pH diagram was performed using the HSC-software (Outotec, Espoo, Finland), as shown in Figure 1, where an increase of temperature from 25 °C to 80 °C did not affect the presence of cobalt ions at different pHs.

As shown in Figure 1, the Pourbaix diagram (potential Eh-pH) of cobalt in a water solution at 25 °C and 80°C confirms the presence of cobalt in the form of $Co^{2+}$ and $Co^{3+}$ at pHs below 0. At an increased potential between 2.0 and 3.0 V, cobalt is available only as $Co^{3+}$.

Regarding the leaching of cobalt, Han and Meng [16] found that the dissolution of cobalt is dependent on diffusion, while the dissolution of divalent oxides is reaction controlled. They reported that the leaching rate of cobalt is generally faster than that of the respective oxides. Huang et al. [8] conducted experiments on the precipitation of cobalt and molybdenum from effluents and used HSC software to compute the potential pH diagrams for a Co-$H_2O$ system at temperatures of 20 °C, 40 °C, 60 °C and 80 °C [8]. Within the parameters of this study, namely, without external potential, Eh = 0, and at pH values close to zero, the stable form of cobalt is a divalent cation within this temperature range, as was observed in our work.

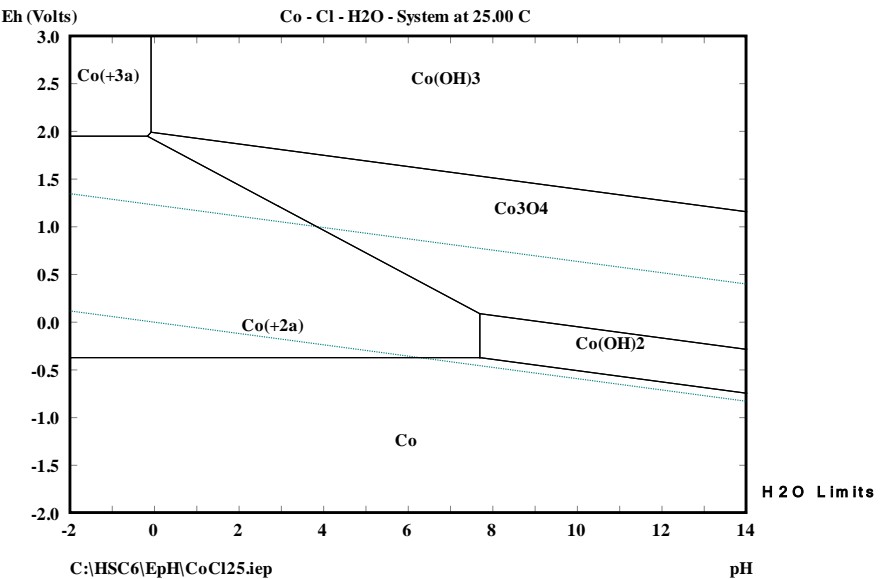

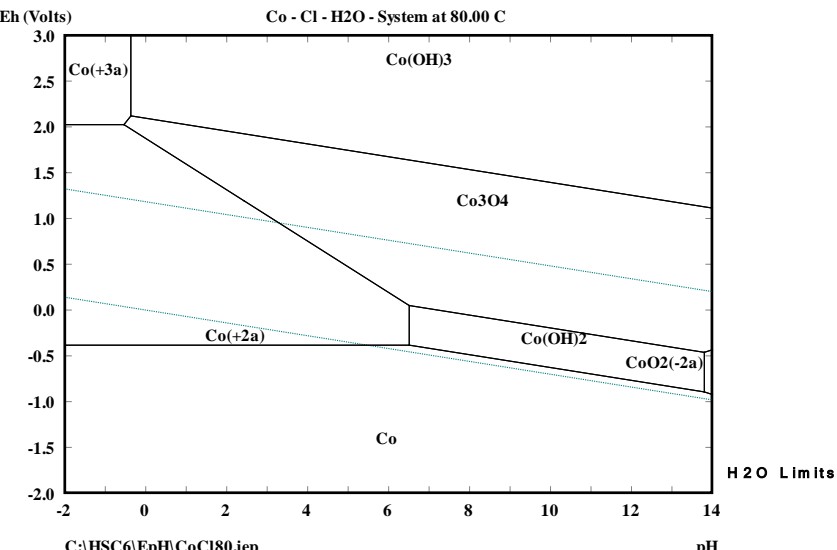

**Figure 1.** eH-pH diagram for Co-Cl-H$_2$O at 25 °C and 80 °C.

## 3. Experiment

This study is focused on polycrystalline diamond (PCD) blanks made by Redies Deutschland GmbH & Co. KG (Aachen, Germany), a manufacturer of wire drawing dies. The material characteristics and procedure are shown in a paper by Kiessling et al. [17]. A SEM (Scanning Electron Microscopy) analysis was performed using a JEOL JSM 7000F Field Emission Scanning Electron Microscope, (2003, JEOL Inc, Peabody, MA, USA) with Energy Dispersive Spectroscopy (EDS), Wavelength Dispersive Spectroscopy (WDS) and Electron back-scattered diffraction (EBSD, JEOL JSM 7000F SEM). The SEM and EDS analysis after polishing the PCD surface is shown in Figure 2 and Table 1.

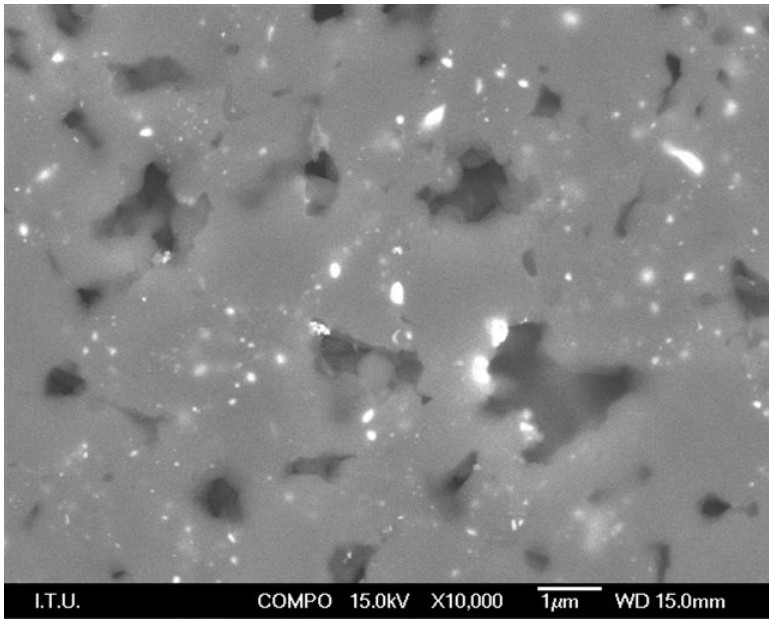

**Figure 2.** SEM image of ground and polished PCD surface, 5 μm class (dark grey areas are the bridged diamond grains with cavities where traces of cobalt also show up as lighter shades).

The maximal value of cobalt in the analyzed sample was 67 wt. %, as shown in Table 1.

**Table 1.** Energy dispersive X-ray spectroscopy image of ground and polished PCD surface, also 5 μm class.

| Content (Weight %) | C | O | Fe | Co |
|---|---|---|---|---|
| Spectrum 1 | 86.81 | 12.94 | - | 0.25 |
| Spectrum 2 | 88.31 | 10.42 | 0.21 | 1.05 |
| Spectrum 3 | 89.35 | 10.32 | - | 0.34 |
| Spectrum 4 | 87.41 | 10.78 | 0.14 | 1.67 |
| Max. | 89.35 | 12.94 | 0.21 | 1.67 |
| Min. | 86.81 | 10.32 | 0.00 | 0.25 |

Different types of samples were used in our work, as shown in Table 2. The columns "volume", "surface area" and "surface-to-volume ratio" contain calculated, and not measured, values.

**Table 2.** Dimensions of PCD samples with grain size of 5 μm.

| Blank Type | Symbol | Diameter [mm] | Height [mm] | Weight [g] | Volume [mm³] | Surface Area [mm²] | Surface/Volume [mm⁻¹] |
|---|---|---|---|---|---|---|---|
| Mant®MSD-14-005 | D14 | 4.05 ± 0.09 | 2.00 ± 0.04 | 0.099 | 25.71 ± 1.40 | 51.14 ± 1.97 | 1.99 (+0.20|−0.18) |
| Mant®MSD-15-005 | D15 | 5.2 | 2.5 | 0.241 | 53.093 | 83.315 | 1.569 |
| Mant®MSD-18-005 | D18 | 5.22 ± 0.02 | 3.50 ± 0.02 | 0.299 | 74.90 ± 0.77 | 100.22 ± 0.68 | 1.338 (±0.023) |

*3.1. Changes in Magnetic Properties of the PCD*

Samples were weighed and measured with a teslametric probe beforehand. During the experiment, liquid samples were taken and a sample of 5 PCD of each type was measured with a teslameter built by Projekt Elektronik GmbH, Berlin, Germany as shown in Figure 3. These ultrasound baths have a nominal frequency of 35 kHz and put out 60 W effectively, while output peaks can occur at up to 240 W. The output level is fixed and the ultrasound irradiation was altered by using it intermittently.

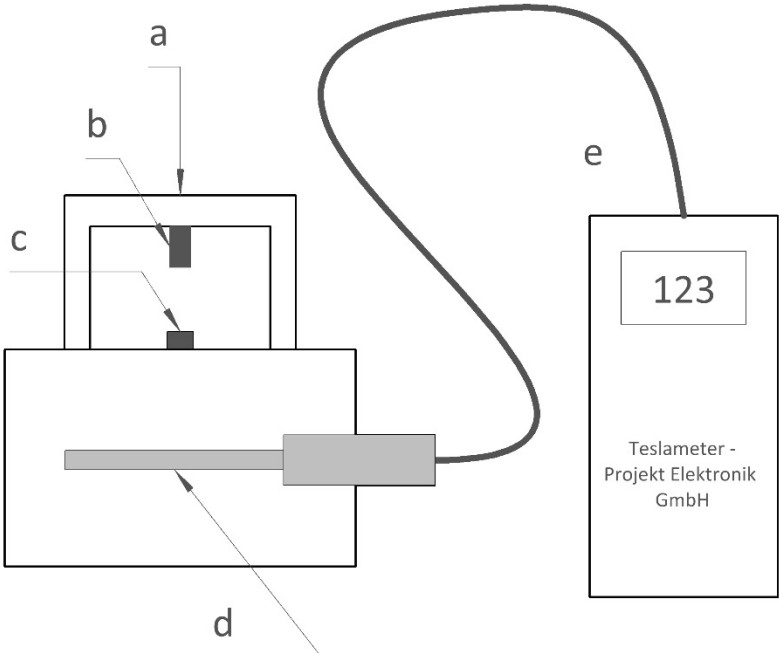

**Figure 3.** Measuring magnetic properties of PCD a: magnet support, b: ∅ 5 mm by 8 mm Nd-Alloy magnet, c: PCD blank, d: probe, e: handheld teslameter.

To investigate the influence of different parameters on the leaching of PCD with aqua regia, the experiments were performed in the following way; see Table 3.

The abbreviations in the first column are successive week number and R1 and R2, representing reactors 1 and 2, respectively. These codes also served as stems for sample identification. The column header S/L is short for solid-to-liquid ratio in units of grams per liter. Constant parameters were stirring speed and batch time. The duration of each batch was between 90 and 100 h. Sample names D14, D15 and D18 are the abbreviated product names of MSD-14-005, MSD-15-005, and MSD-18-005, respectively, all of which are self-supported PCD blanks with diamond grain sizes of 5 μm.

**Table 3.** Parameters for the leaching of cobalt from polycrystalline diamond blank.

| ID | $T_{Bath}$ [K] | PCD Type | Leaching Time [h/d] in the Presence of Ultrasound | S/L [g/L] |
|---|---|---|---|---|
| W1R1 | 333 | D14 | 0 | 15 |
| W1R2 | 353 | D14 | 0 | 15 |
| W2R1 | 333 | D14 + D18 | 0 | 30 |
| W2R2 | 353 | D14 + D18 | 0 | 30 |
| W3R1 | 333 | D14 + D18 | 0 | 45 |
| W3R2 | 353 | D14 + D18 | 0 | 45 |
| W4R1 | 333 | D14 + D18 | 8 h/d | 15 |
| W4R2 | 353 | D14 + D18 | 8 h/d | 15 |
| W5R1 | 333 | D14 + D18 | 8 h/d | 30 |
| W5R2 | 353 | D14 + D18 | 8 h/d | 30 |
| W6R1 | 333 | D15 + D18 | 8 h/d | 45 |
| W6R2 | 353 | D15 + D18 | 8h/d | 45 |
| W7R1 | 333 | D14 + D18 | 24 h/d | 15 |
| W7R2 | 353 | D14 + D18 | 24 h/d | 15 |
| W8R1 | 333 | D14 + D18 | 24 h/d | 30 |
| W8R2 | 353 | D14 + D18 | 24 h/d | 30 |
| W9R1 | 333 | D14 + D18 | 24 h/d | 45 |
| W9R2 | 353 | D14 + D18 | 24 h/d | 45 |

### 3.2. Preparation of Samples

The PCD blanks were weighed and measured with the teslameter beforehand to obtain a '100%' reference value for evaluation. Then, 240 mL of hydrochloric acid and 80 mL nitric acid were prepared in covered beakers for each reactor. Meanwhile, the ultrasound baths had been filled with tap water and their heaters were set to 60 °C and 80 °C to ensure that the bath temperature was nominal at the beginning of the experiment. The gas tightness of the apparatus was checked daily.

As stated, the presence of cobalt in the PCD can be determined electromagnetically. The teslameter consists of a handheld unit and a probe which is sensitive to magnetic field changes of ± 2 mT. To offset any noise or ambient disruptions, a magnetic field from a ⌀ 5 mm by 8 mm rare earth magnet was introduced. The magnet was held at a constant distance to the probe by enclosing the probe in plastic housing and attaching the magnet to a mild steel support structure made of 10 mm × 10 mm square bars.

Data from daily samples was taken and plotted over time. For a better comparison, the measurements were normalized to an initial value of 1, and a reference value without a PCD blank of zero. Then, the daily measurements were plotted as a percentage relative to the initial measurement. For example, in Figure 4, there is the original data plot on the left and the normalized plot on the right.

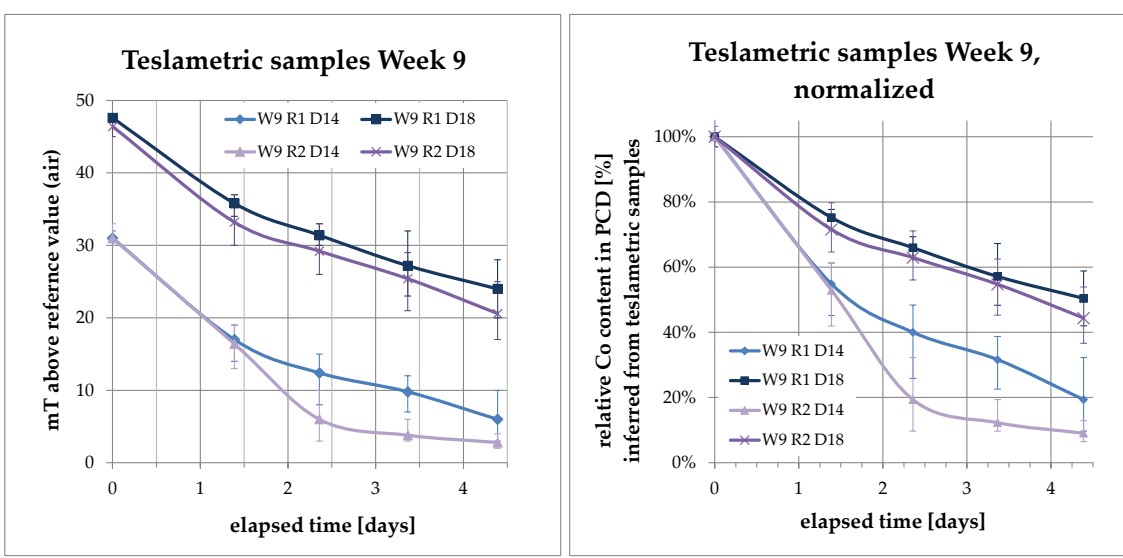

**Figure 4.** Data plot of teslametric samples, (**left**): as measured, (**right**): normalized to initial value.

What's more, this plot can be understood to display the content of ferromagnetic cobalt inside the PCD relative to its initial content. The error bars indicate the extreme values of the sample, while the data points and curve mark the average of the five PCDs sampled per reactor and type.

All data sets have positive curvature, indicating a slowing rate of change over time. The difference in bath temperatures is also visible in this plot, as is the fact that the PCDs in the warmer reactor number 2 were leached to lower values than those in reactor 1.

The effects of ultrasound-assisted leaching on the teslametric measurements were compiled and plotted in order to make a comparison under different conditions and with respect to PCD size, as shown in Figure 5.

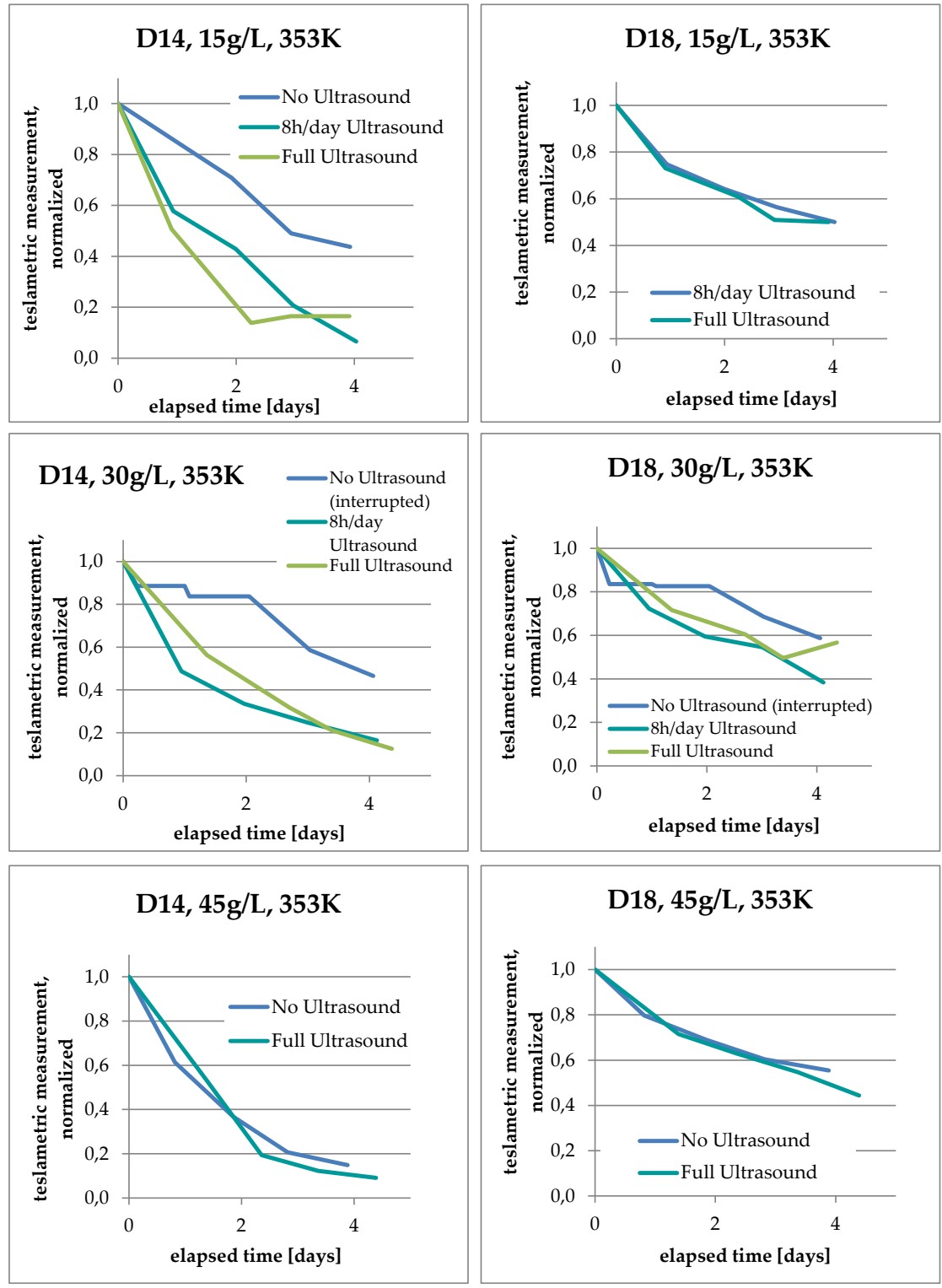

**Figure 5.** Teslametric measurements—comparison of process parameters. "Full Ultrasound" refers to the full-time use of ultrasound.

An additional comparison was made to study the effects of bath temperature with respect to PCD sizes D14 and D18, as shown in Figure 6. At a higher temperature and with full-time ultrasound, the initial drop in the plot is steeper than at lower temperatures and with intermittent ultrasound. When looking at the final values, intermittent ultrasound accomplishes the PCD to achieve lower readings

on the teslameter. The difference between leaching D14 and D18 becomes clear in both images. The decrease of magnetic effects in D18 happens significantly more slowly than in D14, where they seem to reach desaturation within the timeframe. Desaturation in this case means that the remaining magnetic effect is less than 10% of the initial effect. It can be inferred that 90% of the ferromagnetic contents of the PCD have been leached in those cases.

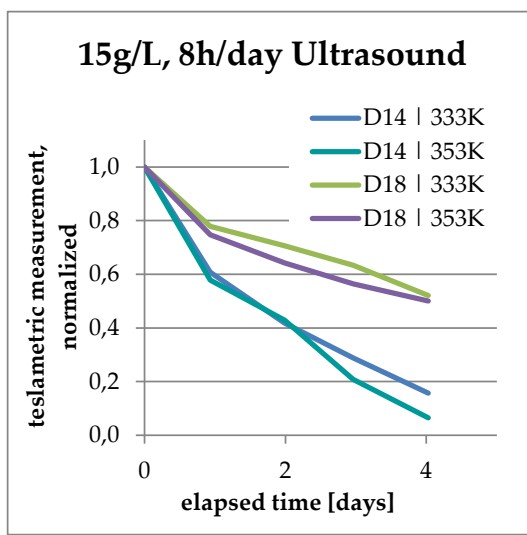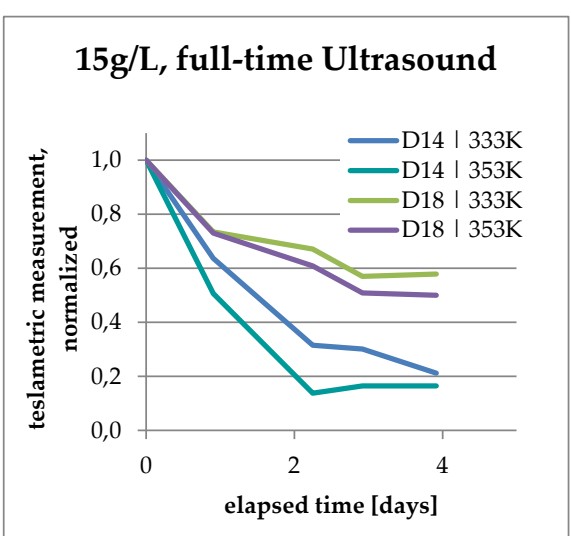

**Figure 6.** Teslametric measurements—comparison with respect to bath temperature

### 3.3. Increase of Co Content in Solution

Considering the analyses of the leaching solutions throughout this study, some trends can be observed. For one, there is the fact that the solution under full ultrasound and at 80 °C accumulates less cobalt than the experiments at 60 °C. This may be due to evaporation losses in the form of hydrochloric acid or water vapor. As stated, the remaining liquid was measured only in weeks 5, 7, 8, and 9. In the extreme case of week nine, in reactor 2, approximately 194 mL remained, which is only 60% of the initial volume. So, the seemingly lower solubility for cobalt at higher temperatures might be due to the reduced available liquid volume and fewer available chlorine anions. Moreover, the formation and escape of nitrous compounds into the off-gas stream is likely to be faster at higher temperatures. The redox potential was measured using a pH meter 7310, InoLAb, WTW, Weilheim, Germany. The redox potential amounted to 460.5 mV for a dissolution of 45 g/L of PCD using full ultrasound. As stated in the literature, aqua regia ensures an oxidation environment. For pH = 0 and Eh = 460.5 mV, cobalt is present as $Co^{2+}$, according to the Eh-pH diagram.

### 3.4. Analysis of Kinetics and Thermochemistry of Cobalt Dissolution

Assuming the PCDs can be viewed as solid particles that lose substance during leaching, according to the unreacted shrinking core model with constant particle size, calculations can be made using two different approaches. First, there are the declining ferromagnetic properties over time, and second, there is the rising cobalt content in solution. Taking equations from the value for X can be calculated in the following way:

$$X_{tesla} = 1 - \left( \frac{\mu_t}{\mu_{t=0}} \right) \tag{10}$$

$$X_{liquid} = 1 - \left( \frac{c_{Co}}{c_{Co,\,final}} \right) \tag{11}$$

where: $\mu$ is the value from the teslametric measurements. $X_{liquid}$ is calculated from the final cobalt concentration in the leaching solution, since a reference is needed and there were no initial analyses of

the cobalt content in the PCD. Furthermore, the expected concentration was surpassed in some weeks, leaving *X* outside the interval {0;1} which would not return realistic results.

Figure 7 shows the respective plots of the diffusion- and reaction-controlled SCM for D14 and D18; these data are exemplary for the weeks denoted in the diagram captions.

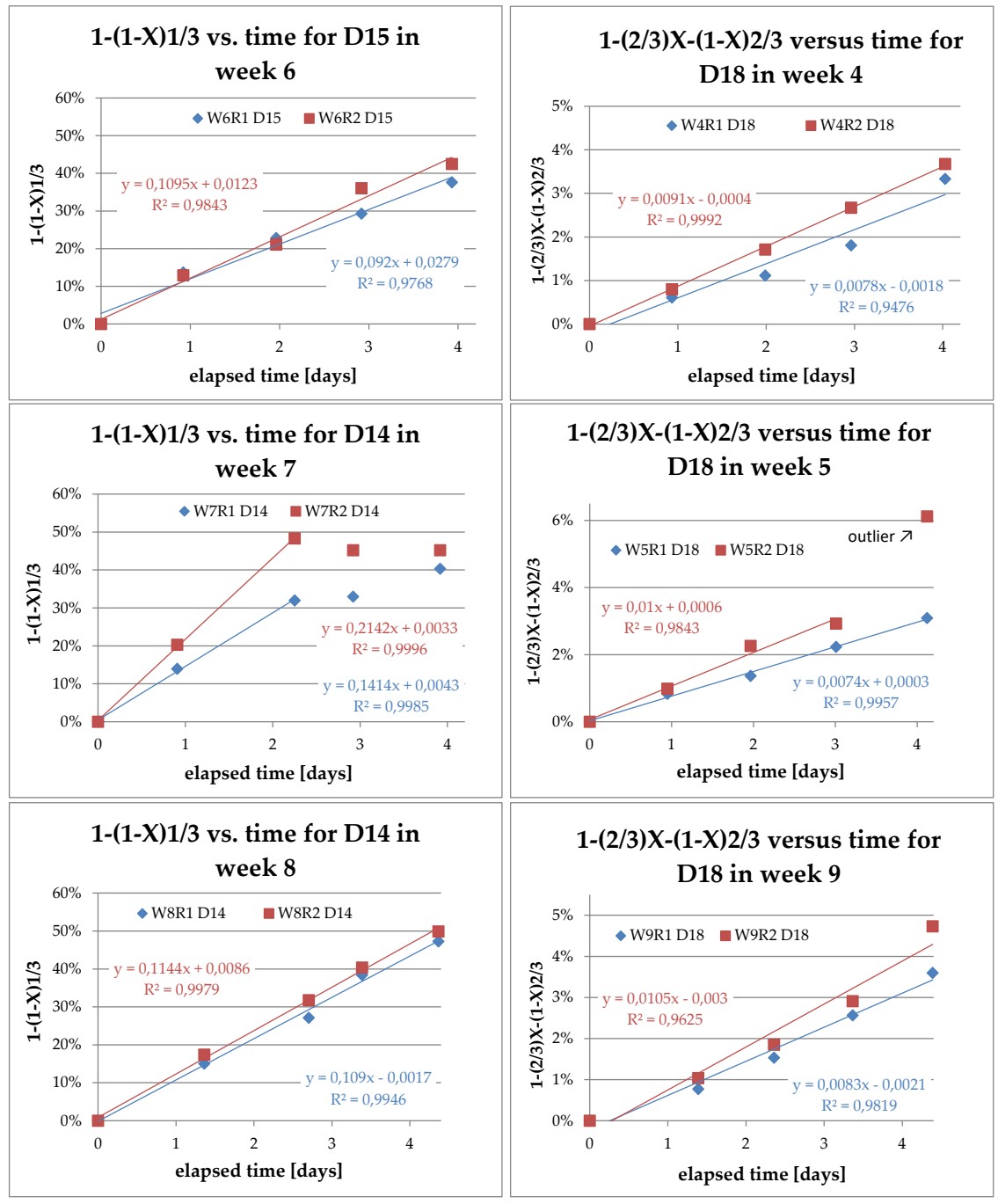

**Figure 7.** SCM plots for D14 and D18, red: 80 °C, blue: 60 °C.

When applying the two models to the teslametric data, it became clear that for D14, the reaction-controlled model was satisfactory, in contrast to D18, which seemed to follow the diffusion-controlled model described by the Ginstling-Brounshtein equation. In all cases, the goodness of the linear regression, $R^2$, was above 0.9, indicating that the fit was satisfactory. In the data from D14 in week 7, there was a very clear change in the reaction rate toward the end of the experiment run. This

is another indicator of either approaching desaturation of ferromagnetic compounds in the PCD or of the saturation with cobalt of the leaching solution. The latter is not very likely, since this observation was made while leaching at a low solid-to-liquid ratio, and the teslametric values for D18 in the same batch were declining at the same time. When the models were applied to the data from the ICP-OES analyses, no linear relationship could be found (as shown in Figure 8). This might indicate that there was a chemical balance prior to the oxidation and dissolution of cobalt, such as the formation of NOCl, which is very important for the subsequent leaching process.

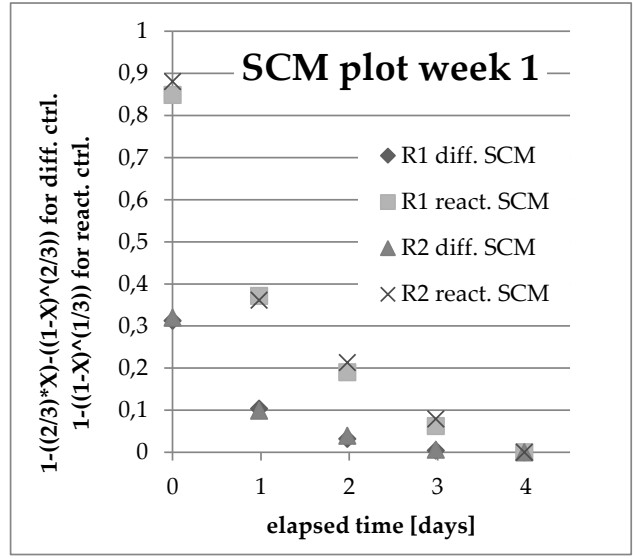

**Figure 8.** SCM plot from liquid samples week 1.

For both D14 and D18, Arrhenius plots were made to see if there were trends concerning the activation energy calculated using Equations (6)–(8). In Figure 9, the results are shown for a select group of experiments for clarity, and to check the extreme cases with regards to the solid-to-liquid ratio and ultrasound irradiation. Leaching D14 at low solid/liquid (S/L) ratios seems to benefit from the use of ultrasound. Temperature, on the other hand, has a bigger effect than ultrasound when leaching at high S/L ratios. For D14, the reaction rates were, in general, higher at 80 °C, as would be expected from a reaction-dependent kinetic model. Unfortunately, the results for D18 were inconclusive. Considering that the experiments were used for orientation and optimization, rather than for a detailed thermochemical analysis, this is an aspect wherein more research is needed.

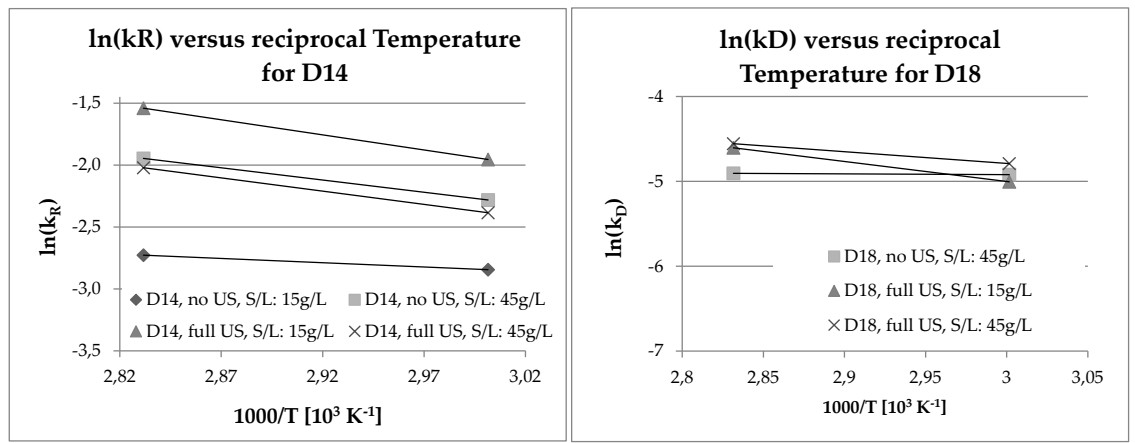

**Figure 9.** Arrhenius plots for D14 and D18.

Regarding the apparent rate constant, other interesting results emerged from the data, as shown in Figure 10. For one thing, there is the question of which particle size, or better, at which depth of penetration into the PCD structure the change from reaction controlled to diffusion controlled SCM might happen. Secondly, looking at the mass balance, the D14 seemed to benefit from intermittent ultrasound just as much as full ultrasound, which should also be recognizable in the rate constant.

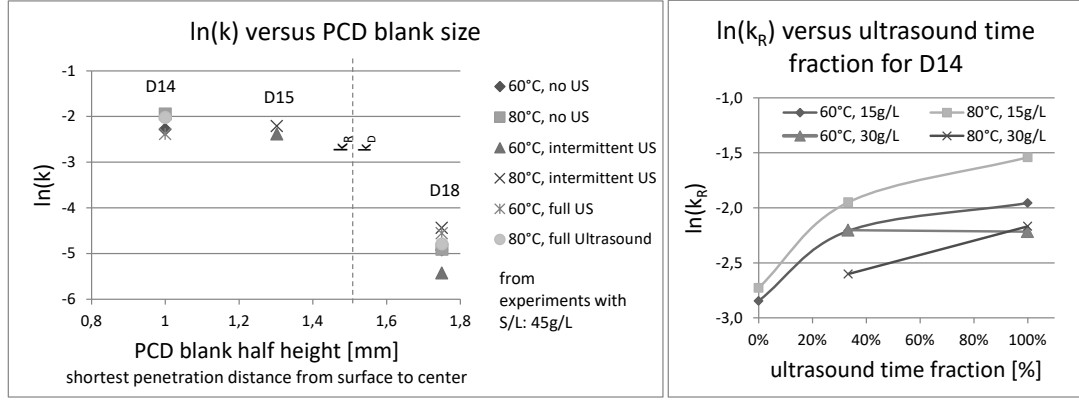

**Figure 10.** Plots of ln(k) over PCD blank size and ultrasound time fraction (D14).

Somewhere in the region of 1.4 to 1.7 mm penetration depth, the leaching of PCD seems to change from the reaction-controlled shrinking core model to diffusion-controlled mechanisms, as described by Equation (6). Moreover, the beneficial effects of ultrasound are reflected in the plot on the right side. For D14, intermittent ultrasound doubles the reaction rate constant and full ultrasound provides a further increase by a factor of 1.5 (see Table 4 below). The maximal obtained activation energy between 60 °C and 80 °C was 20 kJ/mol; below this value, a diffusion controlled-process occurred.

**Table 4.** Activation energy $E_A$ (J/mol) derived from the apparent rate constants.

| Week No. | PCD Type | $\delta \ln(k) = \ln(k_{353K}) - \ln(k_{333K})$ | $\delta/T = (1/353K) - (1/333K)$ | $R_{Ideal}$ [J*K$^{-1}$*mol$^{-1}$] | $E_A = -R_{Ideal}*(\delta\ln(k)/(\delta/T))$ |
|---|---|---|---|---|---|
| 1 | D14 | 0.1183566 | −0.000170142 | 8.314 | 5783.8 |
| 3 | D14 | 0.3347918 | −0.000170142 | 8.314 | 16360.5 |
| 7 | D14 | 0.4153174 | −0.000170142 | 8.314 | 20295.6 |
| 9 | D14 | 0.3621971 | −0.000170142 | 8.314 | 17699.8 |
| 3 | D18 | 0.0136057 | −0.000170142 | 8.314 | 664.9 |
| 7 | D18 | 0.4004776 | −0.000170142 | 8.314 | 19570.4 |
| 9 | D18 | −0.2351197 | −0.000170142 | 8.314 | 11489.8 |

When plotting the calculated Gibbs Energy for the dissolution of cobalt, again, the results for D18 are inconclusive, since there is too little data, and only two different temperatures were investigated. The only statement that could be made is that 80 °C bath temperatures have the tendency to increase $\Delta G^0$ compared to 60 °C. This may again be explained by the more rapid evaporation of the reactants in the warmer reactor. For D14, the same conclusions can be made as with the Arrhenius plot. The lowest $\Delta G^0$ corresponds to a low S/L ratio and the full use of ultrasound in the process, as shown in Figure 11. At higher temperature, the reaction requires less additional energy input.

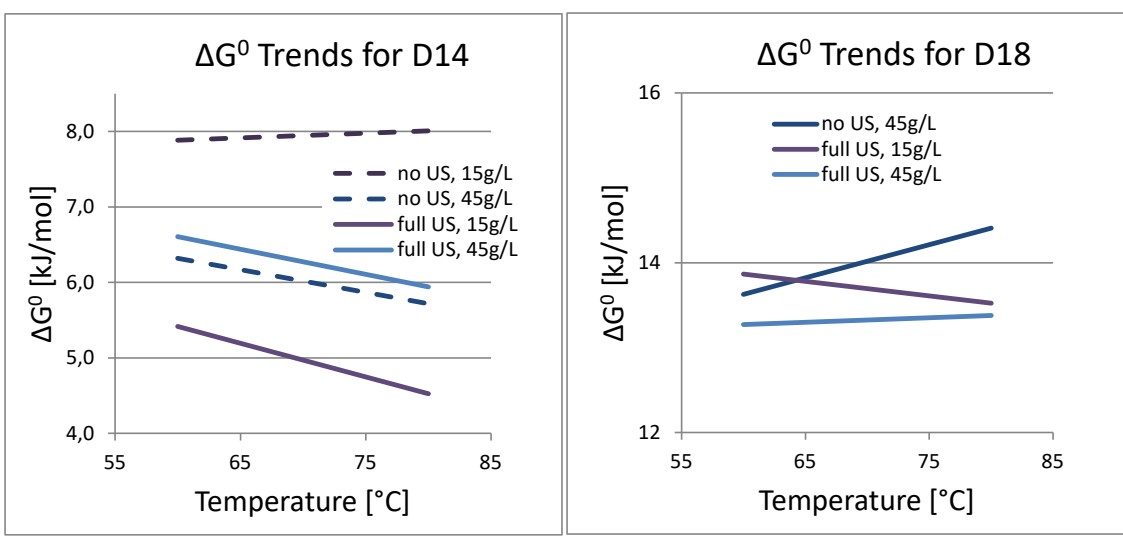

**Figure 11.** $\Delta G^0$ trends for D14 and D18.

For PCD with thicknesses smaller than 2.8 mm, the diffusion-controlled, unreacted shrinking core model according to Equation (6) was confirmed with teslametric data. Similarly, Grénnman et al. [18] used the Ginstling-Brounshtein equation to represent mass transfer across a nonporous product layer. An increase in temperature to above 80 °C does not influence the kinetics of cobalt leaching. Diffusion might be affected by the mixing rate using a mixer and argon gas. Han and Lawson [19] reported a leaching study on metallic cobalt in an acidic medium. This was carried out using a rotating disc geometry to confirm the surface reaction involving hydrogen discharge and the diffusion of hydrogen ions through the boundary layer; together, these phenomena were found to be responsible for the rate of dissolution. The apparent activation energy for cobalt dissolution under the conditions of the experiments was found to be 16.7 kJ/mol, which represents a diffusion-controlled mechanism. Our maximum calculated activation energy amounted 20 kJ for the dissolution of cobalt from PCD blanks, which is in accordance with this.

## 4. Conclusions

This study sought to accelerate the leaching of cobalt from polycrystalline diamond using an ultrasound-assisted leaching process. It seems that varying the ultrasound frequency might be a new strategy to decrease the leaching time for maximal cobalt removal from PCD blanks.

In nine experimental runs of 5 days' duration, cobalt-containing PCD were leached in aqua regia at atmospheric pressure at between 60 °C and 80°. Using two reactors in parallel, temperature, ultrasound irradiation time, solid-to-liquid ratio, and PCD size were varied to determine the optimal parameters and possibly accelerate this process. The redox potential amounted to 460.5 mV for a dissolution of 45 g/L of PCD using full ultrasound, thereby confirming the presence of $Co^{2+}$-ions in solution with a pH value close to zero. Two kinetics models were tested in this work. In both cases, the goodness of the linear regression, $R^2$, was above 0.9, indicating that the fit was satisfactory. Intermittent ultrasound doubles the reaction rate constant, and full ultrasound further increases the rate by a factor of 1.5. The obtained activation energy between 60 °C and 80 °C was 20 kJ/mol, which corresponds to a diffusion-controlled process for dissolution of cobalt. The lowest $\Delta G^0$ corresponds to a low S/L ratio and the full use of ultrasound in the process.

**Author Contributions:** F.K. and S.S. conceptualized and managed the research. S.S. co-wrote the paper. S.G. contributed the SEM and EDS analysis of the PCD surface. B.F. supervised personnel, coordinated resources, and co-wrote the paper. F.K. performed the experiments and wrote the paper. All authors have read and agreed to the published version of the manuscript.

**Funding:** This research was funded by Projektträger (PtJ) Jülich, Grant Number 005-1902-0147.

**Acknowledgments:** We would like to thank to Redies Deutschland GmbH & Co. KG for providing PCD samples as well as additional equipment.

**Conflicts of Interest:** The authors declare no conflict of interest.

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
