# Peer review of "Recovery of Diamond and Cobalt Powders from Polycrystalline Drawing Die Blanks via Ultrasound Assisted Leaching Process—Part 2: Kinetics and Mechanisms"

_metals, doi:10.3390/met10060741_

Round 1

Reviewer 1 Report

Dear authors, I have read your work and want to discuss some controversial points:
1. 72 reaction 5 - CoOs - what is it?
2. 74, 236-238 Co2+ is also present at pH> 0, according to the E-pH diagrams shown.
3. Lines 100-222 are missing somewhere?
4. In many places, there are no dots at the end of sentences.
5. 257 Fig. 5? 259 Fig. 1 or 2?
6. 287, 366 " Error..." ?
7. Poor quality of figures.
8. According to the reviewer, it is incorrect to calculate the activation energy for two temperatures.
9. In General, the work looks unfinished and has many mistakes and inaccuracies. I believe that this material cannot be published in this form.

Author Response

Dear Reviewer,

Thank you very much for your valuable comments and invested time. I put our answers in red color in our letter. I hope that this version can be suitable for the publishing in our Special Issue “Advances in Synthesis of metallic, oxidic and composite powders”.

I am sending our answers:

  1. 72 reaction 5 - CoOs - what is it?

CoOs—(CoO in solid state)

  1. 74, 236-238 Co2+ is also present at pH> 0, according to the E-pH diagrams shown.

Co2+-ion is not only stable form at pH<<1 but also at pH>0.

  1. Lines 100-222 are missing somewhere?

We found it, and changed it. 

  1. In many places, there are no dots at the end of sentences.

We changed it.

  1. 257 Fig. 5? 259 Fig. 1 or 2?

Thank you. We changed the number of Figures.

  1. 287, 366 " Error..." ?

Thank you. We removed it from Text.

  1. Poor quality of figures.

We have removed some figures and injected new figures about our material.

  1. According to the reviewer, it is incorrect to calculate the activation energy for two temperatures.

You have right. It would be better to have 3, 4 or 5 studied temperatures, but we had only two experimental points. The calculated activation energy between 60°C and 80°C confirmed controlled diffusion regime. We put new reference as one explanation

K.N.Han, and F.Lawson, Leaching behaviour of cobalt in acid solutions, Volume 38, Issue 1, October 1974, Pages 19-29

A leaching study of metallic cobalt in an acid medium has been carried out using a rotating disc geometry confirming the surface reaction involving hydrogen discharge and the diffusion of hydrogen ions through the boundary layer. The apparent surface concentration of cobalt in solution is constant irrespective of the disc rotation speed. The cobalt surface concentration is only a very weak function of the bulk hydrogen ion concentration. The apparent activation energy for cobalt dissolution under the conditions of the experiments was found to be 16.7 kJ/mol.

  1. In General, the work looks unfinished and has many mistakes and inaccuracies. I believe that this material cannot be published in this form.

We changed many mistakes and inaccuracies. According to the comments from reviewers we ensured new improved version for the checking. We put new three publications and explanation of our work.

  1. Khawam, A., Flanagan, D., R., Solid state kinetic models- Basics and Mathematical Fundamentals, Phys. Chem. B.2006, 110 (35) 173115-17328
  2. Grénnman, H., Salmi, T., Murzin,, D.. Solid-liquid reaction kinetics – experimental aspects and model development, Rev Chem Eng. 2011, 27, 53–77.  
  3. K.N.Han, and F.Lawson, Leaching behaviour of cobalt in acid solutions, Volume 38, Issue 1, October 1974, Pages 19-29

Reviewer 2 Report

In introduction  a short information on the role of cobalt in PCD manufacturing is missing to explain its presence in the material. Probably it was given in the part I, but as part II is an individual paper should be repeated.

Author Response

Dear Reviewer,

Thank you very much for your valuable comments and invested time. I put our answers in red color. I hope that this version can be suitable for the publishing in our Special Issue “Advances in Synthesis of metallic, oxidic and composite powders”.

I am sending our answers:

Comments and Suggestions for Authors

In introduction a short information on the role of cobalt in PCD manufacturing is missing to explain its presence in the material. Probably it was given in the part I, but as part II is an individual paper should be repeated.

Although we have written it in first part, we repeated this sentence: “Cobalt is used as a solvent catalyst in the production of PCD that would otherwise take even more pressure and a higher temperature to achieve”.

This material contains 20 wt% of cobalt, what is a new concentrate for cobalt recovery, in contrast to mostly used nickel lateritic ore (0.1 wt % Co).

Reviewer 3 Report

The work ''Recovery of diamond and cobalt powders from polycrystalline drawing die scraps via ultrasound assisted leaching process – Part 2: Kinetics and mechanisms'' presents new interesting points, which are valid for metal recovery, circular economy and technological advancement, such as (but not limited to):

- the recovery of cobalt and PC diamond grains from PCD scraps.

- the new method for the direct measure of cobalt leaching from the scraps by teslametric/magnetic measurements. 

However, on the other hand the manuscript should address and report on the following points:

  • More scientific language should be used throughout all the paper.
  • Being part 2 of a previous work, some details needed by the reader, might be present in the first part, but in the current form this second part can't stand as an independent work. Therefore,...

A meaningful characterization of initial PCD samples (e.g. elemental analysis, XRD, SEM) and maybe the characterization of some residues might be implemented for a thorough investigation and evaluation of the metal (and non-metal e.g. polycrystalline diamond)values contained in PCD.

Parallel chemical analyses/characterizations have been performed to validate the magnetic measurements methodology? It is not clear what is the final output of the leaching process, except for the 90% recovery of cobalt, are polycrystalline diamonds clean enough?

More details on ultrasonication parameters should be implemented.  

  • Why aqua regia was used (also in regard to what stated in the manuscript reported below)?

Since ‘’The Pourbaix diagrams by Huang et al. [8] show that the equilibrium for this reaction should be  on the right side of the balance, since the divalent cobalt cation is the stable form at pH<<1 and without external potential.’’

The use of aqua regia has severe environmental concerns and precautionary measurements to be considered e.g. for NOCl, HOCl, H2, NOx. How the authors would address to such point.

  • Was the redox potential measured somehow? It is also mentioned in lines 66-70 that aqua regia provides oxidizing environment. The following statement should be reconsidered.

Within the parameters of this study, namely without external potential, eH = 0, and at pH values close to zero, the stable form of cobalt is the divalent cation within this temperature range, what is situation in our work.

  • Sample names D06, D14, D15 and D18 are abbreviated product names of Mant® 267 MSD-06-005, MSD-14-005, MSD-15-005, and MSD-18-005, all self-supported PCD blanks with 268 diamond grain sizes of 5μm. Please report only the materials used in this paper.
  • May the use of ultrasounds affect the teslametric measurements? What’s the effect of the temperature on the performed magnetic measurements? There is a correction factor applied at the different investigated temperatures? Please provide description in experimental part.
  • Why the model for diffusion control through the fluid film:1− (1−x)2/3=kt was not reported and/or investigated?
  • Applying ultrasound for long times can affect the leaching temperature? Can the temperature variation and ultrasound contributions be distinguished and singularly evaluated in terms of kinetics? there are leaching temperature profiles available?
  • Why SCM plots obtained by ICP and teslametric measurements don’t converge? Knowing the real conc. of Co in the starting material could be beneficial?
  • ''This might indicate that there is a preceding chemical balance prior to oxidation and dissolution of cobalt, such as the formation of NOCl.'' How does NOCl affect the Co leaching kinetics?
  • The activation energies derived from apparent rate are typical of diffusion controlled regime. For D14 the reaction rates in generally were higher at 80°C as would be expected from a reaction dependent kinetic model. Isn’t diffusion affected by the temperature?
  • Line 287 and 366 cross references missing.

Author Response

Dear Reviewer,

Thank you very much for your valuable comments and invested time. I put our answers in red color. I hope that this version can be suitable for the publishing in our Special Issue “ Advances in Synthesis of metallic, oxidic and composite powders”.

Comments and Suggestions for Authors

The work ''Recovery of diamond and cobalt powders from polycrystalline drawing die scraps via ultrasound assisted leaching process – Part 2: Kinetics and mechanisms'' presents new interesting points, which are valid for metal recovery, circular economy and technological advancement, such as (but not limited to):

- the recovery of cobalt and PC diamond grains from PCD scraps.

- the new method for the direct measure of cobalt leaching from the scraps by teslametric/magnetic measurements. 

However, on the other hand the manuscript should address and report on the following points:

  • More scientific language should be used throughout all the paper.

We added some explanation regarding a calculation of activation Energy and comparison with other author: K.N.Han, and F.Lawson, Leaching behaviour of cobalt in acid solutions, Journal of the Less Common Metals, Volume 38, Issue 1, October 1974, Pages 19-29

A leaching study of metallic cobalt in an acid medium has been carried out using a rotating disc geometry confirming the surface reaction involving hydrogen discharge and the diffusion of hydrogen ions through the boundary layer, together, were found to be responsible for the rate of dissolution. The apparent activation energy for cobalt dissolution under the conditions of the experiments was found to be 16.7 kJ/mol, what corresponds a diffusion controlled mechanism. Our calculated activation energy amounts 20 kJ, what is in accordance to this work.

We added two new explanations regarding a study of used kinetics models:

  1. Khawam, A., Flanagan, D., R., Solid state kinetic models- Basics and Mathematical Fundamentals, Phys. Chem. B.2006, 110 (35) 173115-17328
  2. Grénnman, H., Salmi, T., Murzin,, D.. Solid-liquid reaction kinetics – experimental aspects and model development, Rev Chem Eng. 2011, 27, 53–77.  
  • Being part 2 of a previous work, some details needed by the reader, might be present in the first part, but in the current form this second part can't stand as an independent work. Therefore,...

A meaningful characterization of initial PCD samples (e.g. elemental analysis, XRD, SEM) and maybe the characterization of some residues might be implemented for a thorough investigation and evaluation of the metal (and non-metal e.g. polycrystalline diamond) values contained in PCD.

SEM Analysis was performed using JEOL JSM 7000F Field Emission Scanning Electron Microscope, (manufacture year, 2003) with Energy Dispersive Spectroscopy (EDS) and Wavelength Dispersive Spectroscopy (WDS), Electron back-scattered diffraction (EBSDJEOL JSM 7000F SEM.

After polishing of PCD surface, the SEM and EDS analysis is shown in Figure 2. and Table 1.

Figure 2 (attached in my letter). SEM image of ground and polished PCD surface, 5µm class (dark grey areas are the bridged diamond grains with cavities where also traces of cobalt show up in lighter shades)

The maximal value of cobalt in analyzed sample amounts 1. 67 wt %, as shown in Table 1.

Table 1. Energy dispersive X-ray spectroscopy image of ground and polished PCD surface, also 5µm class

Content (Weight %)

C

O

Fe

Co

Spectrum 1

86.81

12.94

0.25

Spectrum 2

88.31

10.42

0.21

1.05

Spectrum 3

89.35

10.32

0.34

Spectrum 4

87.41

10.78

0.14

1.67

Max.

89.35

12.94

0.21

1.67

Min.

86.81

10.32

0.00

0.25

After leaching process, the content of cobalt was analyzed in the solution using ICP OES analysis. Analysis of PCD after leaching was planned, but it was not finished in Corona time in Turkey.

  • Parallel chemical analyses/characterizations have been performed to validate the magnetic measurements methodology?

Parallel chemical analyses (ICP OES)/characterizations have been performed to validate the magnetic measurements methodology

  • It is not clear what is the final output of the leaching process, except for the 90% recovery of cobalt, are polycrystalline diamonds clean enough?

The polycrystalline diamond blanc are clean enough and returned to the production process by Company Redies GmbH Aachen. Unfortunately, the planed SEM and EDS-analysis at the Istanbul Technical University is not finished (Corona time, the sample are lost in Turkey,.)

  • More details on ultrasonication parameters should be implemented.

The ultrasound bath has 60W average ultrasound power. The output level in these Bandelin Sonorex RK 52 H devices is fixed. So the ultrasound irradiation was altered by a) not using ultrasound b) using ultrasound only a third of the experimental time c) by leaving ultrasound running all the time. These ultrasound baths have a nominal frequency of 35kHz and put out 60W effectively while output peaks can happen up to 240W. The output level is fixed and ultrasound irradiation was altered by using it intermittently.

  • Why aqua regia was used (also in regard to what stated in the manuscript reported below)?

From previously performed experiments, the leaching process with aqua regia is still more effective than with organic acid (oxalic acid) despite its harmfulness to the environment. The organic acid is more environmentally friendly, it is a weak acid and do not have the same power to do an oxidation process like aqua regia. We compared leaching with nitric acid in comparison to sulphuric acid and nitric acid in the presence of hydrogen peroxide. Results of the small comparison experiment confirmed nitric acid as a potent leaching agent for cobalt. After four hours the beaker with C-R3 showed some gas bubbles sitting on the PCD while the solution in C-R2 remained visibly unchanged. In the solution of C-R1, however, a pinkish hue could be seen that was stronger near the PCD. After a light shake it was too faint to see anymore. The analytical results from the IME Chemistry Laboratory found the most dissolved cobalt in that beaker (see diagram below)

Figure-Comparison of different leaching solutions. Pure HNO3 dissolved more cobalt than sulfuric acid or aqua Fortis, is shown in my cover  letter

  • The use of aqua regia has severe environmental concerns and precautionary measurements to be considered e.g. for NOCl, HOCl, H2, NOx. How the authors would address to such point.

In our text between Equation 1-5, we have explained the formation of these gases. Especially, a special part was installed for its neutralization using 1- 2M NaOH in our experimental setup

  • Was the redox potential measured somehow?

The redox potential was measured used pH-meter 7310, InoLAb, WTW, Germany. The redox potential amounted 460.5 mV for a dissolution of 45 g/L of PCD using full time of ultrasound.

  • It is also mentioned in lines 66-70 that aqua regia provides oxidizing environment. The following statement should be reconsidered.

Within the parameters of this study, namely without external potential, eH = 0, and at pH values close to zero, the stable form of cobalt is the divalent cation within this temperature range, what is situation in our work.

As known in literature, The aqua regia ensures an oxidation environment. For pH=0 and Eh=460.5 mV, cobalt is present in Co2+according to Eh-pH diagram.

  • Sample names D06, D14, D15 and D18 are abbreviated product names of Mant® 267 MSD-06-005, MSD-14-005, MSD-15-005, and MSD-18-005, all self-supported PCD blanks with 268 diamond grain sizes of 5μm. Please report only the materials used in this paper.

 We removed fractions DO6 from our table. WE used fractions D14, D15 and D18.

Table 2. Dimensions of PCD samples with grain size of 5 µm

Blank type

Symbol

Diameter [mm]

Height [mm]

Weight [g]

Volume [mm3]

Surface area [mm2]

Surface/
Volume
[mm-1]

Mant®
MSD-14-005

D14

4.05
± 0.09

2.00
± 0.04

0.099

25.71
± 1.40

51.14
± 1.97

1.99
(+0.20 | -0.18)

Mant®
MSD-15-005

D15

5.2

2.5

0.241

53.093

83.315

1.569

Mant®
MSD-18-005

D18

5.22
± 0.02

3.50
± 0.02

0.299

74.90
± 0.77

100.22
± 0.68

1.338
( ± 0.023)

  • May the use of ultrasounds affect the teslametric measurements? What’s the effect of the temperature on the performed magnetic measurements? There is a correction factor applied at the different investigated temperatures? Please provide description in experimental part.

An use of Ultrasound does not affect the teslametric measurements. Because of small temperature interval, the change of temperature between 60 and 80 °C did not have an influence on magnetic measurement. No correction factor in this work.

  • Why the model for diffusion control through the fluid film: 1− (1−x)2/3=kt was not reported and/or investigated?

Your proposed Diffusion model (Jander equation, three dimensional”) is included in Equation 6.- Crank-Ginstling- Brounshtein Equation 6- (mass transfer across a nonporous layer), as shown in Table 1 in Reference Henrik Grénnman, Tapio Salmi and Dmitry Yu. Murzin,, Solid-liquid reaction kinetics – experimental aspects and model development , Rev Chem Eng 27 (2011): 53–77                                                           

Equation 6 is the Ginstling-Brounshtein (D4) model. The D4 model is another type of diffusion three-dimensional model in contrast to mostly used Jander model, as reported by Khawan and Flanagan [14]. If a solid particle has a spherical or cubical shape, a contracting sphere/cube model can be applied, as shown with Eq. (7).

  • Applying ultrasound for long times can affect the leaching temperature?

The temperature was not changed for applying ultrasound for long time. The frequency was only 35 kHz. In our previously performed experiments, we used frequency of 1.7-2.5 MHz, but no change of temperature in ultrasound bath.

  • Can the temperature variation and ultrasound contributions be distinguished and singularly evaluated in terms of kinetics?

In this work it was not possible. We used only one frequency 35 KHz, with 60W average ultrasound power and two temperatures 60°C and 80°C. At room temperature no cobalt dissolution. Because of gas formation, in this system with aqua regia an increase of temperature is not environmentally friendly. Our aim was to study an influence of the constant ultrasound frequency in different leaching time (full time and 1/3 used time). Variation of the ultrasound frequency is only possible, if we use new Bandolin devices with other frequency and power, what we did not have.

  • there are leaching temperature profiles available?

Leaching temperature profile is not available.

  • Why SCM plots obtained by ICP and teslametric measurements don’t converge?

This is a good idea. Maybe in our new paper, (3 part) SCM Plot obtained by ICP can be converge with teslametric measurements. I have to point out, that this work is enough innovative in present form and offers one practical solution for company REDIES, GMBH

  • Knowing the real conc. of Co in the starting material could be beneficial?

The real concentration of cobalt is 20 wt % in our sample, what is one concentrate for Cobalt recovery in contrast to 0.1 wt% present in nickel lateritic ore.

  • This might indicate that there is a preceding chemical balance prior to oxidation and dissolution of cobalt, such as the formation of NOCl.'' How does NOCl affect the Co leaching kinetics?

NOCl has a positive influence on the Co-leaching kinetics.

NOCl dissolves in water producing HNO2 and HCl, what are important for the cobalt leaching.

  • The activation energies derived from apparent rate are typical of diffusion controlled regime. For D14 the reaction rates in generally were higher at 80°C as would be expected from a reaction dependent kinetic model. Isn’t diffusion affected by the temperature?

No influence of temperature on diffusion. Diffusion was affected by mixing rate using mixer and argon gas, An increase of temperature above 80°C is not environmentally friendly.

  • Line 287 and 366 cross references missing.

It was mistake. We found it and changed it.

Round 2

Reviewer 1 Report

Dear authors, you have done a good job and corrected the comments made by the reviewers. I wish you success in your further scientific work!

Reviewer 3 Report

The authors addressed to the delicate points of the manuscript. The manuscript still presents small language errors, such as lines 306, 226.

It is suggested to present also the content of the leaching with the other lixiviants e.g. HNO3, etc., along with the manuscript to provide a significant background of the leaching of metallic Co to the reader.